# Biostimulants Application in Horticultural Crops under Abiotic Stress Conditions

**Roberta Bulgari, Giulia Franzoni *** and **Antonio Ferrante**

Department of Agricultural and Environmental Sciences, Università degli Studi di Milano, via Celoria 2, 20133 Milano, Italy; roberta.bulgari@unimi.it (R.B.); antonio.ferrante@unimi.it (A.F.)
* Correspondence: giulia.franzoni@unimi.it; Tel.: +39-02-503-16593

**Abstract:** Abiotic stresses strongly affect plant growth, development, and quality of production; final crop yield can be really compromised if stress occurs in plants' most sensitive phenological phases. Additionally, the increase of crop stress tolerance through genetic improvements requires long breeding programmes and different cultivation environments for crop performance validation. Biostimulants have been proposed as agronomic tools to counteract abiotic stress. Indeed, these products containing bioactive molecules have a beneficial effect on plants and improve their capability to face adverse environmental conditions, acting on primary or secondary metabolism. Many companies are investing in new biostimulant products development and in the identification of the most effective bioactive molecules contained in different kinds of extracts, able to elicit specific plant responses against abiotic stresses. Most of these compounds are unknown and their characterization in term of composition is almost impossible; therefore, they could be classified on the basis of their role in plants. Biostimulants have been generally applied to high-value crops like fruits and vegetables; thus, in this review, we examine and summarise literature on their use on vegetable crops, focusing on their application to counteract the most common environmental stresses.

**Keywords:** plant biostimulant; environmental stress; vegetables

## 1. Abiotic Stresses

Plants are continuously subjected to a multitude of stressful events, from seed germination through to the whole life cycle. These stresses are commonly divided into two categories—biotic and abiotic stresses—depending on the nature of the trigger factor. The first are caused by other living organisms, including insects, bacteria, fungi, and weeds that affect plant development and productivity. The second are generally linked with the climatic, edaphic, and physiographic components of the environment, when they are limiting factors of plant growth and survival. The most important abiotic stresses limiting agricultural productivity, almost all over the world, are drought, salinity, non-optimal temperatures, and low soil fertility. Among these, drought, and nutrient deficiencies are major problems, mostly in developing countries where the incomes of rural people depend on agriculture [1]. Actually, in "The State of Food and Agriculture 2007", FAO reported that only 3.5% of the global land area is not affected by some environmental constraints. In 1982, Boyer estimated that yield losses caused by unfavourable environments were as much as 70% [2,3]. Farooq et al. [4] reported that drought induced a reduction of yield between 13% and 94% in several crops, depending on the intensity and duration of the stress. Afterwards, Cramer et al. [5] estimated the impacts of different abiotic stresses on crop production in terms of the percentage of global land area affected, considering the 2000 and 2007 FAO reports. They also referred to the increasing number of publications focused on this topic between 2001 and 2011. The exact impact of these changes on agricultural systems is extremely difficult to predict and it depends on numerous parameters that are all not always

included in predictive models. Even if some projections show that positive and negative outcomes on crop production could be balanced in the medium term, several studies agree that in the long term, the negative ones will prevail [6,7]. Based on future scenarios, adaptation and mitigation are essential to increase the resilience capacity of agricultural systems and to ensure crops yield and quality. Since environmental conditions cannot be controlled, several strategies on different levels are required, such as agronomical techniques or breeding of more tolerant cultivars [8].

In 2010, at the society's annual conference, Vegetable Breeding and Stress Physiology working groups of the American Society for Horticultural Sciences focused particularly on the "Improvement of Horticultural Crops for Abiotic Stress Tolerance" considering the effects of climate change [9]. Up to now, most studies on climate change impacts focus on major crops, and only few papers pay attention to fruit and vegetable in terms of production, quality, and supply chain [10,11]. An important aspect to take into consideration is the effect of the combination of different stressful factors. Most of the time, crops are subjected to several abiotic stresses that occur simultaneously in the field. In these situations, studying the stresses separately is not enough because plant response is unique and cannot be predicted by the reply obtained when each factor is applied individually [12–14]. Moreover, biotic and abiotic components typically interact in an ecosystem. For instance, environmental conditions affect plant-pest interaction in different ways, by decreasing plant tolerance or increasing the risk of pathogen infection [15,16].

Focusing on horticultural species, the tolerance to abiotic stresses is an important trait because their cash value is usually higher than field crops, they require more resources for farming and because they provide a source of many nutrients, fibre, minerals, and carbohydrates, which are essential in a healthy diet [17]. Food and Agriculture Organization (FAO) reports that about 90% of essential vitamin C and 60% of vitamin A for human comes from vegetables. Indeed, low fruit and vegetable intake is a major contributing risk factor to several widespread and debilitating nutritional diseases. According to the Global Burden of Disease Study, 3.4 million deaths can be attributed to low consumption of fruit and 1.8 million to low vegetables diets worldwide [18]. Therefore, growing high-quality vegetables becomes one of the most important goals of current agriculture, in order to meet the needs of the population and the increasing demand for fruit and vegetables. Abiotic stresses do not only affect the yield but also the quality of these products, triggering morphological, physiological and biochemical changes that can alter the visual appearance and/or the nutraceutical value in a way that the product could become unmarketable [19]. Bisbis et al. [11] investigated the double effects of elevated temperature and increased $CO_2$ on the physiology of different vegetables. They observed several responses according to plant species and severity of the stress, taking into consideration the possible adaptation strategies that could be implemented in order to mitigate the effects of climate change. Nonetheless, these mechanisms are still under-researched and should be studied in depth, because not only different species but different cultivars also could respond differently to the same environmental stress. For example, cultivars with low levels of antioxidants are particularly vulnerable to oxidative stress compared to those with high antioxidant activity [20–23]. This aspect has a particular importance as selection criterion in the choice of appropriate cultivars for a specific situation. Oxidative stress is a common phenomenon caused by several adverse conditions; it generally occurs when the balance between the production of reactive oxygen species (ROS) and the quenching activity is upset by a stressful event [24]. Low levels of ROS are normally produced by different reactions during physiological metabolisms like photosynthesis or respiration, and they play an important signaling role in plant growth and development. Their amount dramatically increases under abiotic stress conditions and, if not controlled could result in cellular damage and death. Besides their toxicity to proteins, lipids or nucleic acids, the increased production of ROS under stressful conditions plays a key role in the complex signaling network of plants stress responses. Their concentration is maintained at non-toxic levels by the activity of the antioxidant system: a wide range of enzymatic or non-enzymatic antioxidant molecules are accumulated in plant tissues to quench ROS induced by stress [25–28]. Moreover, the maintenance of this equilibrium is also dependent on numerous factors, such as the timing of stress application,

its intensity and duration. Indeed, moderate or controlled stress conditions could have a positive effect on quality traits of several crops [29]. For example, water deprivation might be a useful crop management strategy to improve the quality of lettuce and fleshy fruits in terms of nutritive and health-promoting value and taste, by stimulating the secondary metabolism and concentration of different phytochemicals such as α-tocopherol, β-carotene, flavonoid and so on [30,31]. Besides the production of ROS scavenging compounds, plants also increase the biosynthesis and accumulation of compatible solutes with an osmoprotective role, like sugars and proline.

Plants generally reply to non-optimal environmental conditions both with short- and long-term adaptation strategies, by the activation and regulation of the expression of specific stress associated genes [32,33].

Since plants are sessile organisms and they have to cope with adverse external conditions; all these mechanisms are essential for their survival. These strategies are effective if they are activated in time, in order to set a defense response and anticipate the environmental changes that might affect plant growth irreversibly. The trade-off between growth and acclimation metabolisms results in a sort of fitness cost for plants, since energy and nutrients normally destined to growth and production are intended for stress responsive mechanisms [34].

Agronomic management conducted in order to enhance plant tolerance towards abiotic stresses evolved over the centuries due to the technologic progress, climate change, scientific knowledge, and farmers' experiences. The choice of the correct cultivar, the best growing period, the sowing density, and the amount of water or fertilizers are some of the most common strategies applied to mitigate the negative effects of abiotic stresses [8]. Protected cultivation is a cropping technique adopted to preserve plants from unfavourable outdoor conditions. It is mainly suited to vegetables and floriculture production in a non-optimal environment, through the control of temperatures, radiation or atmospheric composition. Another agronomical strategy, especially applied in vegetable crops, is soilless cultivation. This approach allows controlling of water and nutrients, avoiding the use of soil for cultivation and all the problems related to it, like poor quality or contamination.

Grafting is an additional tool adopted to counteract environmental stresses and increase tolerance in vegetable crops. This technique is applied especially to high-yielding fruits and vegetables such as cucurbits and solanaceous to enhance tolerance against saline soil, nutrient or water deficiency, heavy metals or pollutants toxicity [35–37].

Agronomical strategies are essential in mitigating the negative effect of several abiotic stresses, but sometimes their application is not enough. Moreover, current experiments aim to transfer one or more genes involved in signaling or regulatory pathways, or genes encoding to molecules, such as osmolytes and antioxidants, conferring tolerance to a specific abiotic stress [38]. Several functional and regulatory genes involved in abiotic stress tolerance have been identified and studied. Results of these studies can be exploited for genetic improvement aiming to introduce tolerance traits in cultivated crops. Since different physiological traits related to stress tolerance are under multigenic control, the manipulation of a single gene generally is not enough. Hence, scientists have paid more attention to regulatory genes, including transcription factors, due to their ability to regulate a vast array of downstream stress-responsive genes at a time [39–41].

However, the huge existing genetic variability among vegetable species, the lack of knowledge about minor cultivars genome, the complex responses triggered by abiotic stress conditions and the limited strategies currently available make genetic improvement really difficult and often inefficient. Moreover, besides the wide diversity of germplasms available, plant tolerance to stress depends both on stress features such as duration, severity, and frequency, as well as the affected tissues and development stages of crops [24,42–44].

Additionally, the increase of crop tolerance through genetic improvements requires many years of work and different cultivation environments that cannot be always taken into consideration. As a result, several new cultivars that can be used by the growers are released each year.

Another technique widely used for developing stress tolerance in plants is in vitro selection. This culture-based tool allows better understanding of several plants' physiological and biochemical responses to adverse environmental conditions. It has been applied specially to obtain salt/ and drought/tolerant lines in a wide range of plant species, including vegetables [45]. In vitro selection is based on the induction of a genetic variation among cells, tissues or organs, their exposure to a stressor, and the subsequent regeneration of the whole organism starting from the surviving cells [46]. Even if in vitro selection is a less expensive and time-saving approach compared with classic molecular engineering, some limitations, mostly concerning the stability of the selected traits and epigenetic adaptation, still exist.

In addition to these strategies, it has been observed that stress tolerance can also be induced by biostimulants or specific bioactive compounds, if they are applied on vegetable crops when they really need to be protected [47–49]. Biostimulant application on horticultural crops under environmental stress conditions will be discussed in detail below.

## 2. Biostimulants

Biostimulant products have been considered innovative agronomic tools as demonstrated by the increase of scientific publications and by the constant expansion of their market [50]. France, Italy, and Spain are the leading EU countries in the production of biostimulants [51]. According to a new report by Grand View Research, Inc., the biostimulant market size is expected to reach USD 4.14 billion by 2025 [52]. The complex nature of the composition of these products and the wide range of molecules contained makes it complicated to understand and define which compounds are the most active. The isolation and study of a single component is almost impossible and the efficacy of a biostimulant is not due to a single compound but is the consequence of the synergistic action of different bioactive molecules. Moreover, the application rules and time are not always clear. For all these reasons, the European Commission developed a proposal for a new regulatory framework and a draft for a new fertilizer regulation was prepared in 2016. The amendments to the proposal of the European Commission were adopted by the European Parliament in October 2017, while the legislative resolution on the proposal was approved on 27 March 2019 [53–55].

Plant biostimulants are defined as products obtained from different organic or inorganic substances and/or microorganisms, that are able to improve plant growth, productivity and alleviate the negative effects of abiotic stresses [56,57]. Mineral elements, vitamins, amino acids, and poly- and oligosaccharides, trace of natural plant hormones are the most known components. However, it is important to underline that the biostimulant activity must not depend on the product's nutrients or natural plant hormones content. The mechanisms activated by biostimulants are often difficult to identify and are still under investigation [58]. High-throughput phenotyping and omic technologies seem to be useful approaches to understand biostimulants activity and hypothesize a mode of action [59–61]. They can act directly on plant physiology and metabolism by improving soil conditions [62,63]. They are able to modify some molecular processes that allow to improve water and nutrient use efficiency of crops, stimulate plant development, and counteract abiotic stresses [47] by enhancing primary and secondary metabolism [55,61,63].

One of the key points of the discussion is about the application of these products in stressful conditions and their role as nutrients, not with a curative function. In particular, if a product has a direct effect against biotic stresses, it should not be included in the biostimulant category but should be registered as plant protection products.

### 2.1. Classification of Biostimulants in Categories

During the years, different authors have proposed several categorizations of biostimulant products on the basis of their main component or mode of action. In many countries outside the European Union, both kinds of information must be reported on the label in order to register these products [55]. The current classification is based on source of raw material, even if this choice does not always

provide the correct information about the biological activity of the product [56]. Thus, biostimulants are classified as these major groups:

*Humic substances* (*HSs*): they include humic acids, fulvic acids and humins. *HSs* are natural constituents of soil organic matter, resulting from the decomposition processes of plants, animals, and microbial residues, but also from the metabolic activity of soil microbes [57]. It has been observed that treatments with humic substances stimulate plants root growth and development [64,65]. This is reflected in a better uptake of nutrients and water, and enhanced tolerance to environmental stresses, [66,67]. How the *HSs* affect plant physiology is not fully understood. This is due to the molecular complexity of these substances and to the abundance and diversity of plants responses altered by their application. Moreover, a strong relationship between medium properties and *HSs* bioactivity has been reported [68]. The positive effects exerted by these complex aggregates could be ascribed both to the hormone-like activity of some of their component and also to IAA-independent mechanisms [69]. For example, like auxins, *HSs* are able to promote plant growth and induce $H^+$ ATPase activity in plasma membrane [70–72].

*Seaweed extracts*: seaweeds are a vast group of macroscopic, multicellular marine algae that can be brown, red, and green. They are an important source of organic matter and fertilizer nutrients. Seaweed extracts have been used in agriculture as soil conditioners or plant stimulators. They are applied as foliar spray and are able to enhance plant growth, abiotic stresses tolerance, photosynthetic activity, and resistance to fungi, bacteria and virus, improving yield and productivity of several crops [73–75]. Seaweeds used for biostimulant production contain cytokinins and auxins or other hormone-like substances [76]. They also contain many active mineral and organic compounds, including complex polysaccharides such as laminarin, fucoidan, alginates and plant hormones that contribute to plant growth [77]. Recently the potential application of micro-algae as plant biostimulants has been considered [78–80].

*Hydrolysed proteins and amino acids containing products*: hydrolysed proteins are a mixture of amino acids, peptides, polypeptides and denatured proteins that can be obtained by chemical, enzymatic and thermal hydrolysis of proteins (or by combining these different hydrolysis types) from both plant and animal sources [67,81]. Studies reported that the applications of some commercial protein hydrolysate products from animal origin were phytotoxic, having negative effects on plant growth when compared to a commercial protein hydrolysate of plant origin [82,83]. In another study, Botta et al. [84] observed that lettuce plants treated with an animal-based protein hydrolysed had a higher fresh and dry weight compared with the control. Generally, they can induce plant defense responses and increase plant tolerance to many abiotic stresses, as reported by several authors [85–88].

*Microorganisms*: this group includes bacteria, yeast, filamentous fungi, and micro-algae. They are isolated from soil, plants, water, and composted manures or other organic materials. They are applied to soil to increase crop productivity through metabolic activities. They enhance the uptake of nutrients through nitrogen fixation and the solubilization of nutrients, they modify a hormonal status by inducing plant hormones biosynthesis such as auxins, cytokinins, etc.; they also enhance tolerance to abiotic stresses and produce volatile organic compounds (VOCs), which may also have a direct effect on plants. Plant growth-promoting rhizobacteria (PGPR) are able to ameliorate plant responses to abiotic stresses stimulating physical, chemical and biological activities [89,90]. Positive effects are given by microorganisms that form a protective biofilm on root surface enhancing nutrient and water uptake.

Another category of biostimulants includes those derived from extracts of food waste or industrial waste streams, composts and compost extracts, manures, vermicompost, aquaculture residues and waste streams, and sewage treatments among others [91]. Biostimulants derived from agro-industrial by-products were reported to be effective in improving plant productivity, increasing the synthesis of secondary compounds involved in several plant physiological responses, and enhancing the activity of the enzyme phenylalanine ammonia lyase (PAL E.C. 4.3.1.5) [92]. The effect of biostimulant application on PAL activity and on the expression of genes encoding for this enzyme was observed by several authors [56,88,89] and references therein, even if at present it is not possible to define if this is a direct or indirect effect. Because of the diversity of source materials and extraction technologies, the mode of action of these products is not easily determined [55]. The use of by-products as raw material that can be transformed into fertilizing

products is the idea underlying the new fertiliser regulation and the Circular Economy Action Plan, which is focused on reaching a sustainable agriculture. The guidelines for fertiliser regulation, the need to produce in a more environmentally friendly cultivation system maintaining good crop yield and quality, the increase in price of synthetic fertilizer, the withdrawn of several agrochemicals and the multifaceted effects on plants or soil of biostimulants are favouring the expansion of this market.

A new category of biostimulant products, including nanoparticles and nanomaterials, has been recently proposed by Juárez-Maldonado et al. [93]. Nanoparticles and nanomaterials are usually defined as particles with dimensions between about 1 nm and 100 nm that show properties that are not found in their bulk form. They are able to modify the quality of the production and the tolerance to abiotic stresses when applied in small quantities as foliar spray or in nutrient solution, also in vegetable crops [94–97]. Their biostimulant properties seems to be associated with the structure and nature of the materials. The interaction between plant and nanoparticles and nanomaterials surfaces can positively affect ions and metabolites transport and receptors activity by modifying the surrounding environment in terms of energy and charges. This activity is not dependent on chemical composition. Moreover, nanoparticles and nanomaterials release chemical elements like iron or carbon that could be useful for plant when are metabolised.

A study showed that application of zinc oxide nanoparticles on tomato as soil amendment or by foliar spray increased plant height, chlorophyll and total soluble protein content [98].

## 2.2. Effect of Biostimulants on Chlorophyll Content, Photosynthesis and Growth in Vegetables

Biostimulants can be used in vegetable cultivation to improve productivity and yield, and to enhance plant health and tolerance to stress factors. Indeed, they have positive effects on plant metabolism, both in optimal and sub-optimal environmental conditions.

Many authors have observed that plant based biostimulants and seaweed extracts often increase the colour of leaves by stimulating chlorophyll biosynthesis or reducing its degradation [99,100]. Leaf colour is an important quality parameter in vegetable crops because it contributes to the visual appearance of the product, especially in leafy vegetables for which the greenness influences the consumer's appeal. In addition, a higher chlorophyll content also allows for a greater photosynthetic activity of leaves. High concentration of leaf pigments (chlorophyll and carotenoids) has been observed after biostimulant treatments in rocket [101,102], in lettuce, and endive by Bulgari et al. [103]. Amino acids or seaweed extract application had positive effects on photosynthetic pigments, P and K content, fresh and dry weight of celeriac leaves [104]. Similar results have been observed after root inoculation with several plant growth promoting bacteria (PGPR) in broccoli (*Brassica oleracea* 'italica') using *Bacillus cereus*, *Brevibacillus reuszeri*, and *Rhizobium rubi* [105], and tomato under non-stressful conditions treated with PGPRs belonging to the genera *Bacillus*, *Pseudomonas* and *Azotobacter* [106], in strawberry (*Fragaria ananassa*) with five PGPRs (*Bacillus subtilis*, *Bacillus atrophaeus*, *Bacillus spharicus* subgroup, *Staphylococcus kloosii*, and *Kocuria erythromyxa*) [107] and also in lettuce grown under salt stress after inoculation with *Serratia* sp., *Rhizobium* sp., and *Azospirillum* [108,109]. Brown seaweeds are widely used as a biostimulant products to improve plant growth, and recently a phenolic compound isolated from *Ecklonia maxima* showed stimulatory effects in cabbage plants, improving photosynthetic pigments concentration, phytochemicals and myrosinase activity [110].

Abdalla [111] reported that moringa leaf extracts increased vegetative growth, chlorophyll content, total sugars, phenols, ascorbic acid, and photosynthetic rate of rocket salad. Similar effects have been observed in fennel [112,113] and squash under water stress condition (plants under a deficit irrigation of 80% or 60% ETc) [114]. In tomato plants it led to a greater fruit weight, volume and firmness, and enhanced titratableacidity, chlorophyll and ascorbic acid content [115].

Luziatelli et al. [116] recently found that different vegetal-derived bioactive compounds significantly increased the chlorophyll content and fresh weight of lettuce. Kulkarni et al. [117] investigated the promoting effect of bioactive molecules derived from smoke and seaweed in spinach

and they observed that morphological, physiological and biochemical parameters including growth, chlorophyll and carotenoids content were positively improved.

Broccoli plants were significantly affected by two different products: Goemar BM86 and Seasol. The content of micro- and macro-nutrients increased, and also the leaf area, stem diameter and biomass, as reported by Gajc-Wolska et al. [74] and Mattner et al. [118].

Paradiković et al. [119] studied the effect of four different commercial biostimulants (Radifarm, Megafol, Viva, and Benefit), containing amino acid, polysaccharides and organic acids as active compounds on pepper plants and observed an increase in both yield and fruit quality. Radifarm and Viva treatments also affected tomato plants, stimulating the root apparatus in optimal and drought condition, respectively [120,121].

Recently, a sago bagasse hydrolysate was tested on tomato plants. The product showed a growth promoting ability as observed by the higher seed germination and protein and sugar content compared to the control. Moreover, the expression of the genes related to carbon and nitrogen metabolisms increased [122].

### 2.3. Biostimulants and Crop Tolerance to Abiotic Stresses

Table 1 is a summary of biostimulant products or bioactive molecules from different origins that have been evaluated for amelioration of abiotic stresses in several vegetables species. The biostimulants effectiveness to counteract the stressful condition depends on several factors, such as timing of application and their mode of action. The application of biostimulants can be carried out with different timings: before the stress affects the cultivation, during the stress, or even after. They could be applied on seeds, when plants are in early stages of growth, or when crops are fully developed, depending on the desired results [123]. As general consideration, biostimulants that contain anti-stress compounds, such as proline or glutamic acid, can be applied when the stress occurs or during stress conditions. On the contrary, those that are involved in the activation of bioactive compounds biosynthesis must be applied before the stress occurs. Proper timing of application during crop development differs from species to species and it also depends on the most critical phases for crop productivity. Thus, the identification of the right time of biostimulant application is as important as the determination of the exact dose, in order to avoid waste of product, high production costs, and unexpected results. Biostimulants can be applied as foliar spray or to the roots, at sowing for protecting the seedling in the early development stages, in a floating system nutrient solution or during blooming or fruit setting. There is no general recipe that works for a crop species and in each stress situation.

The protective role of biostimulants on plants has been increasingly studied. These products are able to counteract environmental stress such as water deficit, soil salinization, and exposure to sub-optimal growth temperatures in several ways [47,56,124,125]: They improve plant performance, enhance plant growth and productivity, interact with several processes involved in plant responses to stress, and increase the accumulation of antioxidant compounds that allow decrease in plant stress sensitivity.

More recent results of interest on vegetable crops tolerance have been obtained after the application of different exogenous treatments. Cao et al. [126] reported that a lower red to far-red ration improved tomato seedling tolerance to salt stress, acting on phytochrome activity. Mertinez et al. [127] showed positive results obtained after the application of exogenous melatonin in tomato plants grown under a combination of salinity and heat. Another interesting approach to induce tolerance to abiotic stresses is soaking plant seeds with different compounds, synthetic or natural. This strategy is generally called seed priming and has been deeply reviewed by Asharaf et al. [128].

### 2.3.1. Biostimulants and Cold or Chilling Stress

Low temperatures reduce plant metabolism and delay physiological responses. A reduced metabolism, consequent to cold stress, leads to an inhibition of the activity of photosystem II, called photoinhibition. Cold induces damages to cell membranes with destabilization of the phospholipid layers.

In tomato, cold tolerance has been enhanced by the application of psychrotolerant soil bacteria. Several strains have been isolated from soil during winter conditions and used as a cold protectant.

Tomato treated with these psychrotolerant bacteria showed higher seeds germination, reduced membrane damage, and antioxidant systems activation when exposed to chilling temperatures [129,130]. These soil bacteria can be considered as putative biostimulants for protecting plants against cold stress. Since low temperature causes stress to plant, especially during transplant, Marfà et al. [131] studied the effect of an enzymatic hydrolysates obtained from animal haemoglobin on strawberry plants in the firsts growing stages. They observed an increase in roots biomass and in the early production of fruit. The same product was also tested on lettuce plants subjected to cold stress and an increase in fresh weight, dry weight, specific leaf area, and relative growth rate was observed [132].

External applications of an amino acid biostimulant (Terra-Sorb® Foliar) on lettuce plants grown in different cold situations led to an increase in fresh weight and to an higher stomatal conductance [84]. A typical plants response to stress is the accumulation of compatible osmolytes, such as amino acids, which confer tolerance. The exogenous application of amino acids has the benefit of avoiding protein breakdown and saving energy resources in plants, even if the exact mechanism of action is not fully understood. Pepper (*Capsicum annuum*) seedlings were treated with 5-aminolevulinic acid in order to improve chilling tolerance through three different methods—soaking the seeds, spraying the leaves or drenching the soil. All the applications showed good effects in terms of stress tolerance. Fresh biomass, proline, sucrose, and water content were significantly higher while membrane permeability was reduced [133].

Positive effects on coriander plant grown in cold vegetative chambers have been observed in response to Asahi SL or Goemar Gateo (Arysta Life Science) treatments [124]. Results obtained by the study of stress indicators such as antioxidant activity, photosynthetic pigment concentration and activity, hydrogen peroxide and malondialdehyde amount showed that biostimulant application affected different metabolic pathways in a positive way, leading stressed plants to a phase of acclimation to low temperature. The biostimulant action against cold stress usually increases the accumulation of osmotic molecules by stimulating the biosynthetic pathways that lead to the cold protectant substances. These biostimulants also increase membrane thermostability, reducing the chilling injury.

### 2.3.2. Biostimulants and Heat Stress

Global warming and the projection of a rising temperature have a negative impact on agriculture [134,135]. High temperatures could induce several damages to plant cells, disturbing proteins synthesis and activity, inactivating enzymes and damaging membranes. The range between 30 °C and 45 °C is the optimal temperature for structural integrity and enzymal activity, which are irreversibly denatured when temperature increases above 60 °C. As a consequence, physiological activities like photosynthesis or respiration are affected. An overproduction of toxic compounds, like reactive oxygen species, causing oxidative stress, is one of the most frequent throwbacks [136]. As response, plants start synthesizing compatibles solutes in order to maintain cell homeostasis and turgor, organize proteins, and cellular structures. Moreover, they generally close stomata and increase the number of trachomatous, in order to prevent water loss. Also, at the molecular level there is a variation of the expression of genes involved in the synthesis or activity of antioxidant enzymes related to ROS scavenging, osmolytes or transporters. Temperature above optimum inhibits seeds germination and retards plant growth. Heat stress could negatively affect the yield by interfering with the reproductive phase, decreasing pollen vitality and germination, inhibiting flower differentiation and development and reducing fruit set, which ultimately reduces growth and yield.

Tomato is considered one of the most sensitive species to non-optimal temperatures, and heat stress often results in long style lengths and in a decreased fruit set [137]. There is little information in the literature about treatments specifically applied to vegetable crops exclusively against high temperature since, most of the time, heat stress is combined with drought or salinity. The application of brassinosteroids on tomato [138] and snap bean [139] has resulted in a higher biomass accumulation and net photosynthesis rate, increased growth and quality of snap bean pod in terms of NPK content and the total free amino acids levels in leaves. This might be due to the protective role of brassinosteroids

on the photosynthetic apparatus from oxidative stress, increasing the ability to regenerate RuBP and carboxylation efficiency.

Nahar et al. [140] investigated the effect of exogenous application of glutathione against heat stress. Mung bean seedlings treated before their exposition to high temperature, showed a reduced oxidative stress and methylglyoxal content, a reactive compound that damages cells. This results in a more efficient antioxidant defense system. Pre-treatment with glutathione enhanced tolerance to short-term heat stress, improving plant physiological adaptation. For example, leaf relative water content and turgidity, which usually decreases under high temperature, were protected. Positive effect on mung bean has been observed in response to the application of nitric oxide [141] and ascorbic acid [142]. Nitric oxide treatment resulted in a promotion of photosynthetic activity, increasing the quantum maximum efficiency of PS2. It also affected electrolyte leakage, leading to a better cell membrane integrity. Oxidative stress, lipid peroxidation, and $H_2O_2$ content were decreased and antioxidant enzyme activity was restored. Similar results have been obtained after the application of proline and abscisic acid on chickpea [143,144]. Chickpea is sensitive to high temperature that generally leads to yield and quality losses. After treatments, membrane damage, measured as electrolyte leakage, MDA and $H_2O_2$ levels was decreased, while leaf water content was increased. These effects might be related with the osmoprotectant role of proline and with the accumulation of osmolytes after ABA treatments. Treated plants also showed a high chlorophyll content and this result, which has already been seen in other experiment with exogenous proline, could be related to membrane stability. The activity of oxidative metabolism was enhanced in treated plants, as expected also by the less oxidative damage of cells.

As discussed above, melatonin treatment exerts a positive effect to counteract chilling stress in coriander plants; otherwise, Martinetz et al. [127] found that melatonin treatments also have a protective role against the combination of heat and salt stress in tomato plants. Biostimulant treatments used against heat stress protect cell membranes by increasing their stability and reduce or avoid the accumulation of ROS.

### 2.3.3. Biostimulants and Salinity Stress

Among abiotic stresses, salinity is one of the main damaging factors affecting plant growth and metabolism as an effect of osmotic stress caused by salt. Sodium chloride (NaCl) is the more abundant salt presents in saline environments and is toxic in higher concentrations [145]. It happens especially near the coasts, where crops are frequently irrigated with saline water [85,146]. In many Mediterranean areas, the problem of seawater intrusion may cause a reduction of 50% of yield in lettuce cultivation, as reported by Miceli et al. [147]. A significant reduction of both fresh weight and chlorophyll content is a typical effect of salinity condition on plants and was observed also in spinach [148], in bean [149] and other crops [150]. Besides, chlorophyll content is a central parameter of the product quality particularly in green leafy vegetable, not only in terms of plant physiology status but also from a market point of view. This is a huge problem for vegetable crops where the edible parts are leaves, sprouts or flower buds. Consumers choices, in fact, are guided mostly by the visual appearance of products, hence a less green leafy vegetable or a malformed fruit are generally not accepted.

Salt stress causes a nutrient imbalance due to the limited uptake of the nutrients from the soil, threatening the nutritional quality of horticultural crops. Nutrient availability is compromised by salinity that causes several disorders such as competitive uptake with other ions like $Ca^{2+}$, P and K, mobility problems within the plant and a reduced water potential [151–155]. The solubility of micronutrients such as Cu, Fe, Mn, Mo and Zn is also affected by the pH of the soil solution, and in saline condition their availability is very low. Bano et al. [156] reported an important reduction of total phenolics, total soluble proteins and a suppressed activity of catalase, superoxide dismutase and peroxidase in carrot under saline condition. Salt stress could also alter several metabolic processes in plants, such as photosynthesis [157,158], respiration [159], phytohormone regulation, protein biosynthesis, nitrogen assimilation [160], and can also generate secondary oxidative stress [146,161]. It generally leads to a decrease of production and to a lower quality of the final product, due to an

inhibition of leaves and roots growth and a change in leaf colour [17]. To verify the effects deriving from the applications of biostimulants, several trials on lettuce plants under salt stress were performed, since this crop is considered moderately sensitive to salinity.

Lucini et al. [85] showed that a plant-derived protein hydrolysate improved tolerance to salinity in lettuce plants, increasing yield and dry weight. Treated plants also have a higher performance and an increased maximum quantum efficiency of PS2 compared to the control. Similar results have been recently observed in lettuce plants in response to the application of an organic commercial biostimulant named Retrosal® [162].

Several experiments have been carried out using different PGPR that are able to enhance abiotic stress tolerance. Inoculation with *Azospirillum brasilense* showed positive results on lettuce [163,164], sweet pepper [165], chickpea and faba beans [166] grown under salty environment. Lettuce fresh weight, dry weight, ascorbic acid content, and germination percentage were increased; also, the visual appearance of the final product was better because of higher chlorophyll levels. In chickpeas and faba beans, the inoculation relieved the stress caused by salinity, increasing the root and shoot growth compared with the non-inoculated plants. Sweet pepper is a salt-sensitive crop and inoculation showed positive effect mitigating deleterious effects of NaCl. Dry weight, indeed, was higher than non-inoculated plants under several salt concentrations. Moreover, the inoculation also increased the $CO_2$ assimilation rate. A similar result has been obtained by Cordovilla et al. [167] applying two different *Rhizobium* strain on faba bean and pea plants. Pea plants inoculated with tolerant strain showed no reduction by salt stress condition in shoot and roots dry weight. The same strain was, however, not effective on faba beans. These results highlight the variation existing inter and intra species, and the difficulty in improving tolerance through selection and breeding. A comparable experiment has been carried out by Mayak et al. [168] on tomato seedling. They tested several strains of *rhizobacterium* and found that plants inoculated with *Achromobacter piechaudii* and irrigated with saline water had a higher fresh and dry weights and an increased water use efficiency. Yildirim et al. [169] obtained similar results in squash with the application of several biological products based on the *Bacillus* and *Trichoderma* species.

It is known that humic acids have a lot of beneficial effect stimulating shoot and root growth and improving environmental stress tolerance even if the exact mechanism of action is not completely clear. These activities were confirmed in several vegetable crops like sweet pepper [170], beans [171] and cucumber [172] grown under different salt stress conditions.

Bioactive compounds present in seaweed extracts are able to improve plant tolerance against abiotic stresses too. Two seaweed-based plant biostimulants containing *Ascophyllum nodosum* named Super Fifty® and Acadian were applied respectively on lettuce [173] and strawberry [174] and were associated with a significant increase in yield and root dry weight, despite the adverse salinity condition.

Sulphated exopolysaccharides extracted from the microalgae *Dunaliella salina* were applied on tomato plants to investigate their potential effect alleviating salt stress damages. Results obtained showed that treatment enhance plant growth, antioxidant enzymes activities and several metabolic mechanisms related to jasmonic acid pathway [175].

The application of seaweed extracts from *Sargassum muticum* and *Jania rubens* significantly alleviated the negative effects of salt through regulation of amino acids metabolism, ionic content balanced and improved antioxidant defence in chickpeas plants. Amino acids such as serine, threonine, proline and aspartic acid were identified in roots as responsible for salt stress amelioration [176].

Besides lettuce and pepper, bean is also considered a salt sensitive plant but in most developing countries it is cultivated in saline conditions. Several plant extracts based on licorice root, *Moringa oleifera* or maize grain have been tested on common bean by Egyptian researchers [177–181]. They observed that soaking seeds in propolis or maize grain extract improves seed germination percentage, stability of cell membrane and relative water potential under saline conditions. Antioxidant system activity was increased while lipid peroxidation and electrolyte leakage were reduced compared with the control plants. *Moringa oleifera* leaf extract, used alone or in combination with salicylic acid, and administered

as foliar spray or as seed soaking, improved several physiochemical parameters as chlorophyll and carotenoids concentration, total soluble sugars and ascorbic acid content. A very similar trial has been carried out with licorice root extract and best results have been recorded integrating seed soaking and foliar spray applications.

A recent study highlighted the ability of a bee-honey based biostimulant to improve the tolerance of onion plants to salinity stress. Indeed, treated plants showed higher biomass, bulb yield, and photosynthetic pigments. Moreover, the osmoprotectans content as proline, soluble sugars and total free amino acids, the membrane stability index and the enzymatic and non-enzymatic antioxidant activity were enhanced [182]. Hence, biostimulants applied in case of salinity stress induce the accumulation of osmolytes, in order to enhance the cell osmotic potential and the level of protective molecules against oxidative stress.

### 2.3.4. Biostimulants and Drought Stress

Abiotic stresses are closely connected with the problem of resources availability and farmers are frequently forced to work in suboptimal conditions. A more sustainable use of resources also concerns water availability, a critical growing factor. The increasing use of aquifer-based irrigation by farmers worldwide poses a serious threat to the long-term sustainability of the agricultural system. Over-utilization of this dwindling water supply is leading to an ever-enlarging area in which productive farming itself has ceased or is threatened. Moreover, the increase of irrigation leads to a higher risk of soil salinization. Scientists generally agree with the perspective that several regions could become arid due to the negative impacts of global climate change on water resources [183]. Since one of the main effects of biostimulants is to improve water use efficiency, their application could be a possible strategy to reduce the amount of water added to crops [184]. Drought stress strongly influences plant gas exchange changing photosynthetic and transpiration rates, which are directly linked to yield. Application of *Ascophyllum nodosum* on broccoli [185] and spinach [186] enhanced gas exchange through the reduction of stomatal closure, resulting in increased plant resistance to water stress. Leaf yellowing is another common symptom of drought stress due to chlorophyll degradation during leaf senescence and is used as reliable indicator of metabolic and energetic imbalance in plants under stress. Biostimulant treatments with *A. nodosum* increased total chlorophyll content in tomato leaves [187]. A reduction of water loss, wilting damages and 3-carbon dialdehyde MDA after biostimulant applications were observed. Similar results have been obtained by Petrozza et al. [188] in responses to Megafol treatments in tomato plants. The results revealed that treated plants were healthier than non-treated ones in terms of biomass and chlorophyll fluorescence. Moreover, plants treated with the biostimulant product were able to recover more quickly when they had access to water. The expression of two drought stress marker genes was analysed and the results obtained showed that treated plants were experiencing a low level of water stress.

Sometimes, water stress in plants is caused by bacterial infection clogging xylem vessels and preventing water flow. Romero et al. [189] demonstrated that treatments with *Azospirillum brasilense*, a strain isolated in arid environments, delayed wilting of tomato plants. Treated plants, indeed, showed a high xylem vessels area, resulting in a more efficient water transport from the soil to the leaves. On the other hand, there are several strains of bacteria populating soil promoting plant growth through its metabolic activities and plant interactions. They produce exopolysaccharides, phytohormones, 1-aminocyclopropane-1-carboxylate (ACC) deaminase, volatile compounds, inducing several metabolic plant responses as accumulation of osmolytes and antioxidants, or up or down regulation of stress responsive genes and alteration in root morphology leading to a tolerance of water stress [190,191]. Some examples are reported below. Tomato seedlings treated with *Achromobacter piechaudii* were stimulated to accumulate biomass during the stress period and, the amount of ethylene that usually has negative effects on membrane status was lower than control [168].

Arshad et al. [192] investigated the growth of two plants promoting rhizobacteria on pea (*Pisum sativum*) crop grown under drought stress condition in different phenological phases. They observed

that PGPR containing ACC-deaminase, a precursor of ethylene, significantly decreased the stress effects on growth and yield too. Positive results in terms of antioxidant and photosynthetic pigments activity have been collected in basil plants treated with *Pseudomonas sp.* under water stress conditions [193].

Seaweed extracts are already largely used for cultivated plant treatments and most of them contain plant growth hormones, auxins, abscisic acid, cytokinins, gibberellins, polyamines, oligosaccharides, betaines and brassinosteroids. A micro-algae-based biostimulant with known composition was tested on water stressed tomato plants. Results revealed that biostimulant application reduced the damaging effects of stress, increased plant height, root length, and enhanced the number and area of the leaves [78]. Biostimulants are capable of reducing drought injures, are able to enhance the biosynthesis of osmolytes and antioxidants against ROS, such as observed for salinity stress, and of plant hormones, like abscisic acid, regulating transpiration and avoiding excessive water losses.

### 2.3.5. Biostimulants and Nutrient Deficiency

One of the roles ascribed to biostimulant products is the ability to increase nutrient uptake [53] through different strategies. For instance, they are able to change soil structure or nutrient solubility, modify roots morphology directly or ameliorate nutrient transport in plants [194]. Their application might be really useful in poor soil conditions and in low input horticultural cultivation systems [195]. Indeed, soil nutrient imbalance is an increasing problem for farmers that spend a lot of money every year on fertilizers to resume soil fertility. All these mechanisms result in better nutrient use efficiency for both micro- and micro-nutrients.

Several experiments have been performed to investigate if the application of biostimulants allows a reduction of fertilizers without affecting crop yield and quality.

Koleška et al. [196] showed that the application of a biostimulant product named Viva® on tomato plants, growing under reduced NPK nutrition, help counteract the negative effects of nutrient deficiency. For example, lycopene and chlorophyll content that is usually affected by the availability of macronutrients was preserved in treated plants grown with NPK reduction. Moreover, biostimulant application helped maintain cell homeostasis and prevent oxidative stress. A similar experiment was performed by Anjum et al. [197] on garlic plants grown with half of the recommended dose of nutrients. Garlic growth and yield were positively affected by the biostimulant application in combination with a low dose of macronutrients.

A seaweed-based product (Kelpak®) has been tested on okra seedlings grown with different nutrient deficiencies [198]. Treatments were applied three times a week and were compared with a polyamine solution treatment. Plants treated with the biostimulant showed an increase in growth parameters, such as shoot length, stem thickness, leaves and roots numbers, and fresh weight under phosphorous and potassium deficiency. Kelpak® efficacy might be due to the combination of auxins, cytokinins and polyamines contained in the product.

Spinelli et al. [199] measured the effects of another commercial seaweed extract, named Actiwave® on the vegetative and productive performance of strawberry plants grown on an iron deficient substrate. They found that vegetative growth, chlorophyll content, stomatal density and photosynthetic rate were enhanced after biostimulant treatment. Fruit production and weight were also increased. Nutrient uptake might have been positively influenced by the more developed root system of treated plants. Treatment also contrasted the negative effects of iron chlorosis and this could be linked to betaine contained in this product.

The positive effects of seaweed extracts are usually ascribed to their polysaccharide content that helps the soil structure; nevertheless, Vernieri et al. [102] obtained good results by applying Actiwave in a hydroponic system with different concentrations of nutrient solutions. Yield and leaf area were higher in rocket plants grown with the lowest nutrient concentration, indicating a better nutrient use efficiency.

Most of the biostimulant contains a mixture of different amino acids and short peptides that are usually called protein hydrolysates. They have a positive effect on plant growth and protection against several stresses. The Cerdán et al. [200] study showed that amino acids origin might influence

the efficacy of the product. Tomato plants grown under iron deficiency conditions and treated with two products containing amino acids from plant and animal origin showed different responses. Plant-derived amino acids promoted growth and chlorophyll content both in controlled and iron deficiency conditions. This effect might be ascribed to glutamic acid content. Indeed, this amino acid plays an important role in nitrogen metabolism [201] and chlorophyll biosynthesis [202].

Nutrient imbalance might be the cause of several disorders during plant growth and development. Blossom-end rot in pepper is usually caused by a local calcium deficiency in young fruits. Parađiković et al. [203] tested four different biostimulant products for their effects on yield and BER incidence on pepper. They also evaluated the application as foliar spray or in a nutrient solution of the same products. The results obtained revealed that biostimulants applications helped to reduce the occurrence of BER and increase yield. Moreover, nutrient accumulation in fruits and leaves was promoted by the treatments.

These experiments revealed that biostimulant products cannot totally replace fertilizers but could be really useful to reduce the amount of mineral nutrition or help in nutrient deficiency and imbalanced situations. For example, in the floating system cultivation of baby leaf such as rocket, the nutrient solution can be reduced by 75% of Hoagland's solution [101].

The biostimulants that help reduce nutrient deficiencies usually improve crops nutrient uptake by increasing root biomass, nutrient transport/translocation, and enzyme activities involved in nutrient assimilation.

## 3. Conclusions and Future Prospects

This review reports the progress on the recent development of biostimulant products with special emphasis on their effects, improving tolerance to abiotic stresses in vegetable crops. During their life cycle, crops are often exposed to abiotic stresses, acting individually or in combination, which could dramatically reduce the yield and quality of products. Biostimulants could represent an effective and sustainable tool to enhance plant growth and productiveness, improving tolerance against abiotic stresses. In fact, biostimulants have been successfully applied for:

- improving nutrients and water use efficiency of crops;
- enhancing tolerance against salinity, water stress, cold, high temperature, etc.;
- increasing yield and quality of agricultural crops.

It is important to consider that the complex and variable nature of raw materials used for their production and the heterogeneous mixture of components of the final product can make it difficult to attribute a specific mode of action to each biostimulant. The situation is further complicated by the high number of plants, bacteria and in general, substances included into the category of plant biostimulants. For example, two products obtained by two different plants would fall in the same category, but their effects and their mode of action might be completely different. Moreover, the opposite situation may occur; the same product may produce different effects when applied on different plants. This could be related to the genetic variability among species, variety or cultivars. In addition, the biostimulant activity of a product may also depend on the nature and severity of the abiotic stress.

It must also be considered that trying to link a specific mode of action only to the main component of a product might be a mistake because it would be like excluding the effect of the molecules that are presents in small quantities or in traces, but it is known that the efficacy of biostimulant products is the result of a synergistic or antagonistic effect of many components. Furthermore, our understanding of the mode of action also depends on the amount of information provided by scientific papers, on the numbers of analyses performed, and on their investigation level. The availability of innovative research tools will surely improve the knowledge of biostimulant composition, but this information will not be exhaustive. Therefore, the biostimulant mode of action can be understood through plant responses at the physiological, biochemical, and molecular levels.

**Table 1.** Examples of biostimulant products or substances with a biostimulant effect on horticultural crops to counteract abiotic stress conditions.

| ABIOTIC STRESS | SEVERITY AND TIME OF E×POUSURE | BIOSTIMULANT PRODUCT OR SUBSTANCES WITH A BIOSTIMULANT EFFECT | DOSE | APPLICATION METHODS AND NUMBER OF TREATMENTS | CROP | BENEFICIAL EFFECTS | REFERENCE |
|---|---|---|---|---|---|---|---|
| Chilling or cold stress | 6 °C for 6 days | Asahi SL (Sodium para-nitrophenolate, sodium ortho-nitrophenolate, sodium 5-nitroguaiacolate) / Goëmar Goteo (Composition (w/v): organic substances 1.3–2.4%, phosphorus ($P_2O_5$). 24.8%, potassium ($K_2O$) .4.75%) | 0.1% | Foliar spray (3×) | *Coriandrum sativum L.* | ↓electrolyte leakage ↑Chlorophyll *a* and carotenoids ↑Fv/Fm ↑E ↑gs ↓Ci | [124] |
| | 10, 12 °C for 7 days / 15 °C for 7, 10 days | *Flavobacterium glaciei, Pseudomonas frederiksbergensis, Pseudomonas vancouverensis* | - | Seed inoculation | *Solanum lycopersicum* | ↑shoot height ↑root length ↑biomass accumulation ↓electrolyte leakage ↓lipid peroxidation ↑proline accumulation ↑SOD, CAT, APX, POD, GR activity | [129,130] |
| | −6 °C for 5 nights | Pepton 85/16 (enzymatic hydrolysates obtained from animal haemoglobin. L-α amino acids (84.83%) and free amino acids (16.52%), organic-nitrogen content (12%), mineral-nitrogen content (1.4%), potassium content (4.45%), iron content (4061 ppm), very low heavy-metal content) | 2 L ha$^{-1}$, 4 L ha$^{-1}$ | Injection into the soil (5×) | *Fragaria × ananassa* | ↑new roots ↑flowering ↑fruit weight | [131] |
| | −3 °C for 4 h | Pepton 85/16 | 0.4, 0.8, 1.6 g L$^{-1}$ | Soil application (1×) | *Lactuca sativa L.* | ↑fresh and dry weight ↑SLA ↑RGR | [132] |
| | 4 °C for 8 days or nights /6 °C for 8 days only to the roots | Terra-Sorb® Foliar (Free amino acids (ASP, SER, GLU, GLY, HIS, ARG, THR, ALA, PRO, CIS, TYR, VAL, MET, LYS, ILE, LEU, PHE, TRP) 9,3% (w/w), Total amino acids 12% (w/w), Total nitrogen (N) 2,1% (w/w), Organic Nitrogen (N) 2,1% (w/w), Boron (B) 0,02% (w/w), Manganese (Mn) 0,05% (w/w), Zinc (Zn) 0,07% (w/w), Organic matter 14,8% (w/w)) | 3 mL L$^{-1}$ | Foliar spray (3×) | *Lactuca sativa L. var. capitata* | ↑roots fresh weight ↑green cover % | [84] |
| | 3 °C for 48 h | 5-aminolevulinic acid | 0, 1, 10, 25, 50 ppm (15 mL for seed soaking and 25 mL for soil drench) | Seed soaking/ foliar spray/soil drench (1×) | *Capsicum annuum* | ↓visual injuring ↑chlorophyll ↑RWC ↑gs ↓membrane permeability ↑shoot and root mass ↑SOD activity | [133] |

Table 1. *Cont.*

| ABIOTIC STRESS | SEVERITY AND TIME OF E×POUSURE | BIOSTIMULANT PRODUCT OR SUBSTANCES WITH A BIOSTIMULANT EFFECT | DOSE | APPLICATION METHODS AND NUMBER OF TREATMENTS | CROP | BENEFICIAL EFFECTS | REFERENCE |
|---|---|---|---|---|---|---|---|
| Drought stress | Occlusion of xylem vessels | *Azospirillum brasilense (BNM65)* | - | Seed inoculation | *Solanum lycopersicum* | ↑height plants ↑dry weight ↑xylem vessel area | [189] |
| | No irrigation for 5 days | Megafol® (Composition (w/v): total nitrogen (N) 3.0% (36.6 g L$^{-1}$); organic nitrogen (N) 1.0% (12.2 g L$^{-1}$); ureic nitrogen (N) 2.0% (24.4 g L$^{-1}$); potassium oxide (K$_2$O) soluble in water 8.0% (97.6 g); organic carbon (C) of biological origin 9.0% (109.8 g L$^{-1}$)) | 2 mL L$^{-1}$ | Foliar spray (1×) | *Solanum lycopersicum* | ↑leaf area ↑RLWC | [188] |
| | 50% ET | *Ascophyllum nodosum* | 0.50% | Foliar spray and drench | *Spinacia oleracea* | ↑RLWC ↑leaf area ↑fresh and dry weight ↑SLA ↑gas exchange | [186] |
| | No irrigation until symptoms of wilting appear | Pseudomonas spp. (*P. putida P. fluorescens*) | - | Seed inoculation | *Pisum sativum* | ↑grain yield ↑root growth ↑shoot length ↑number of pods per plant ↑chlorophyll | [192] |
| | No irrigation for 12 days | *Achromobacter piechaudii* (ARV8) | - | Seedling inoculation | *Solanum lycopersicum* | ↑fresh and dry weight of seedling ↑plant growth ↓ethylene | [168] |
| | No irrigation for 12 days | *Achromobacter piechaudii* (ARV8) | - | Seedling inoculation | *Capsicum annuum* | ↑ fresh and dry weight of seedling ↑plant growth | [168] |
| | No irrigation for 7 days | *Ascophyllum nodosum* | 0.33% | Foliar spray (2×) | *Solanum lycopersicum* | ↑RWC ↑plant growth ↑foliar density ↑chlorophyll ↓lipid peroxidation ↑proline ↑soluble sugars | [187] |
| | No irrigation for 2 days | *Ascophyllum nodosum* + amino acids | - | Soil application (1×)/ foliar spray (3×) | *Brassica oleracea var. italica* | ↑Pn ↑gs ↑chlorophyll | [185] |
| | 40, 70% field capacity | Gibbrellic acid and titanium dioxide | 250, 500 ppm (GA3) 0.01, 0.03% (titanium nanoparticles) | Stems and foliar spray (2×) | *Ocimum basilicum* | ↑CAT activity ↓lipid peroxidation ↑LRWC | [95] |
| | No irrigation | VIVA® | - | 2× | *Solanum lycopersicum* | ↑plant biomass ↑roots biomass | [120] |
| | 60, 40% field capacity | *Pseudomonades, Bacillus lentus, Azospirillum brasilens* | - | Seed inoculation | *Ocimum basilicum* | ↑CAT, GPX activity ↑chlorophyll | [193] |
| | 60, 40% ET | Moringa leaf extract | 3% | Foliar spray (2×) | *Cucurbita pepo* | ↑growth ↑HI ↑WUE ↑Fv/Fm ↑PI ↑soluble sugars ↑free proline ↓electrolyte leakage ↑membrane stability | [114] |

**Table 1.** *Cont.*

| ABIOTIC STRESS | SEVERITY AND TIME OF EXPOUSURE | BIOSTIMULANT PRODUCT OR SUBSTANCES WITH A BIOSTIMULANT EFFECT | DOSE | APPLICATION METHODS AND NUMBER OF TREATMENTS | CROP | BENEFICIAL EFFECTS | REFERENCE |
|---|---|---|---|---|---|---|---|
| | 35 °C | Nano-TiO$_2$ | 0.05, 0.1, 0.2 g L$^{-1}$ | Foliar spray (1×) | *Solanum lycopersicum* | ↑gs ↑E ↑ Pn | [94] |
| | 40/30 °C for 8 days | Brassinosteroids | 0.01, 0.1, and 1.0 mg L$^{-1}$ | Foliar spray (1×) | *Solanum lycopersicum* | ↑antioxidant enzyme activities ↓H$_2$O$_2$ ↓MDA ↑shoot weight | [138] |
| | 35.2 °C (Tmax) | Brassinosteroids | 25, 50, 100 ppm | Foliar spray (2×) | *Phaseolus vulgaris* | ↑plant length ↑number of leaves, branches and shoots per plant ↑fresh and dry weight ↑pod weight ↑N, P, K in bean pods | [139] |
| | 45 °C for 90 min | Nitric oxide | 150 μM | Immersion of leaf disks | *Phaseolus radiatus* | ↑Fm ↓electrolyte leakage | [141] |
| **Heat stress** | 35/25 40/30 45/35 °C | Ascorbic acid | 50 μM | In a nutrient solution | *Phaseolus radiatus* | ↑% germination ↑seedling growth ↓electrolyte leakage ↑TTC reduction ability ↑RLWC ↓MDA ↓H$_2$O$_2$ ↑antioxidant activity ↑ascorbic acid ↑GSH ↑proline | [142] |
| | 35/25 40/30 45/35 °C | Proline | 5, 10, 15 μM | In a nutrient solution | *Cicer arietinum* | ↑% germination ↑shoot and root length ↓electrolyte leakage ↑chlorophyll ↑RLWC ↓lipid peroxidation ↓H$_2$O$_2$ ↑GSH ↑proline | [143] |
| | 35/25 40/30 45/35 °C for 10 days | Abscisic acid | 2.5 μM | In a nutrient solution | *Cicer arietinum* | ↑shoot length ↑osmolytes ↑chlorophyll ↑cellular oxidizing ability | [144] |
| | 42 °C for 48 h | Glutathione | 0.5 mM | - | *Vigna radiata L.* | ↑RLWC ↑chlorophyll ↑proline ↓MDA ↓ H$_2$O$_2$ ↓O$_2^-$ ↓LOX activity ↑ascorbate ↓GSSG | [140] |
| **Heat and salt stress** | 35 °C and 75 mM NaCl for 15 days | Melatonin | 100 μM | Foliar spray (5×) | *Solanum lycopersicum* | ↑biomass ↑Pn ↑gs ↑E ↑chlorophyll a ↑carotenoids ↑Fv/Fm ↑efficiency of PSII ↑ETR ↑antioxidant capacity ↓H$_2$O$_2$ ↓lipid peroxidation ↓protein oxidation | [127] |

**Table 1.** *Cont.*

| ABIOTIC STRESS | SEVERITY AND TIME OF E×POUSURE | BIOSTIMULANT PRODUCT OR SUBSTANCES WITH A BIOSTIMULANT EFFECT | DOSE | APPLICATION METHODS AND NUMBER OF TREATMENTS | CROP | BENEFICIAL EFFECTS | REFERENCE |
|---|---|---|---|---|---|---|---|
| **Iron deficiency** | - | Actiwave® (*Ascophyllum nodosum*)(Composition (w/v): total nitrogen (N) 3.0% (38.7 g L$^{-1}$); organic nitrogen (N) 1.0% (12.9 g L$^{-1}$); ureic nitrogen (N) 2.0% (25.8 g L$^{-1}$); potassium oxide (K$_2$O) soluble in water 7.0% (90.3 g L$^{-1}$); organic carbon (C) of biological origin 12% (154.8 g L$^{-1}$); iron (Fe) soluble in water 0.5% (6.45 g L$^{-1}$); iron (Fe) chelated by ethylenediaminedi (2-hydroxy-5-sulfophenylacetic) acid (EDDHSA) 0.5% (6.45 g L$^{-1}$); zinc (Zn) soluble in water 0.08% (1.03 g L$^{-1}$); zinc (Zn) chelated by Ethylenediaminetetraacetic acid (EDTA) 0.08% (1.03 g L$^{-1}$)) | 10 mL in 20 mL tap water | In a nutrient solution | *Fragaria ananassa* | ↑vegetative growth ↑chlorophyll ↑stomatal density ↑photosynthetic rate ↑ fruit production ↑berry weight | [199] |
| | - | Amino acids | 0.1, 0.2 mL L$^{-1}$ / 0.2, 0.7 mL L$^{-1}$ | Root application/foliar spray (4×) | *Solanum lycopersicum* | ↑plant growth ↑root and leaf ferrum chelate reductase activity ↑chlorophyll ↑leaf Fe ↑Fe$_2$:Fe ratio | [200] |
| **Reduced NPK** | NPK reduced of 40% | VIVA® (Composition (w/v): total nitrogen (N) 3.0% (37.2 g L$^{-1}$); organic nitrogen (N) 1.0% (12.4 g L$^{-1}$); ureic nitrogen (N) 2.0% (24.8 g L$^{-1}$); potassium oxide (K$_2$O) soluble in water 8.0% (99.2 g L$^{-1}$); organic carbon (C) of biological origin 8.0% (99.2 g L$^{-1}$); iron (Fe) soluble in water 0.02% (0.25 g L$^{-1}$); iron (Fe) chelated by EDDHSA 0.02% (0.25 g L$^{-1}$)) | 10.5 mL /plant | Foliar spray | *Solanum lycopersicum* | ↑yield ↑ascorbic acid ↑lycopene ↑chlorophyll ↑carotenoids | [196] |
| | NPK deprivation | Kelpak (*Ecklonia maxima,* containing polyamine, cytokinins and auxins, putrescine, spermine ) | 0.40% | In a nutrient solution (twice per week for 8 weeks) | *Abelmoschus esculentus* | ↑number of leaves ↑number of roots ↑stem thickness ↑shoot weight ↑root weight ↑leaf area | [198] |
| | NPK reduced of 50% | Bio-Cozyme (concentrated micro-biological biostimulant and soil inoculants. Total Nitrogen (N) 0.20%, Soluble Potash (KO) 5.00%, Magnesium (Mg) 1.40%, Boron (B) 0.20%, Copper (Cu) 0.50%, Iron (Fe) 3.00%, Manganese (Mn)1.00%, Molybdenum (Mo) 0.0.25%, Zinc (Zn) 2.00%, Humic Acid, humates & derivatives 8.00%, Vitamins, E, C, B Complex, organic acids, natural sugars carbohydrates, amino acids 1.40%) | 2 kg ha$^{-1}$ | Foliar application (4×) | *Allium sativum* | ↑bulb yield ↑plant height ↑NPK in leaves | [197] |

Table 1. *Cont.*

| ABIOTIC STRESS | SEVERITY AND TIME OF E×POUSURE | BIOSTIMULANT PRODUCT OR SUBSTANCES WITH A BIOSTIMULANT EFFECT | DOSE | APPLICATION METHODS AND NUMBER OF TREATMENTS | CROP | BENEFICIAL EFFECTS | REFERENCE |
|---|---|---|---|---|---|---|---|
| Salt stress | 30, 50, 80 mol m$^{-3}$ NaCl for 30 days / 40, 80, 120 mol m$^{-3}$ NaCl | *Azospirillum brasilense* | - | Seed inoculation | *Lactuca sativa* | ↑germination % ↑total fresh and dry weight ↑biomass partition ↑plantlets number ↑plantlets dry weight ↑total leaf fresh weight ↑leaf area ↑leaves number ↑chlorophyll ↑root dry weigh ↑ascorbic acid ↑plant survival after transplant | [163,164] |
| | 40, 80, 120 mM NaCl | *Azospirillum brasilense/Pantoea dispersa* | - | Inoculation | *Capsicum annuum* | ↑plant dry weight ↑K$^+$:Na$^+$ ratio ↑gs ↑relative growth rate ↑net assimilation rate ↓ Cl$^-$ accumulation ↑NO$_3^-$ concentration ↑CO$_2$ assimilation | [165] |
| | 714 mg·L$^{-1}$ NaCl | *Azospirillum brasilense* (ATCC 29,729) | - | Soil inoculation | *Cicer arietinum* | ↑nodule formation ↑shoot dry weight | [166] |
| | 100 mmol L$^{-1}$ NaCl | *Rhizobium leguminosarum* (GRA19–GRL19) | - | Seedling inoculation | *Vicia faba / Pisum sativum* | ↑plant growth | [167] |
| | 50, 100 mM NaCl | *Bacillus species, Bacillus pumilis, Trichoderma harzannum, Paenibacillus azotoformans and polymyxa* | - | Seed treatment/ watering | *Cucurbita pepo* | ↑fresh weight ↑potassium uptake ↓sodium uptake ↑ K$^+$:Na$^+$ ratio | [169] |
| | 30, 60, 120 mM (NaCl, Na$_2$SO$_4$, CaCl$_2$, CaSO$_4$, KCl, K$_2$SO$_4$, MgCl$_2$, MgSO$_4$) for 60 days | Humic acid | 0.05, 0.1% | Soil application | *Phaseolus vulgaris* | ↑plant nitrate, nitrogen and phosphorus ↓soil electricity conductivity ↓proline ↓electrolyte leakage ↑plant root and shoot dry weight | [171] |
| | - | Acadian (*Ascophyllum nodosum*) | - | Soil application | *Fragaria ananassa* | ↑yield ↑growth ↑root length ↑surface area, volume and number of tips ↑numbers of crowns | [174] |
| | 80 mM NaCl | Super Fifty® (*Ascophyllum nodosum*) | 0.4, 1, 2.5, 10 mL L$^{-1}$ | In the nutrient solution | *Lactuca sativa* | ↑root, stem, total plant weight | [173] |
| | 25 mM NaCl | Protein hydrolysates | 2.5 mL L$^{-1}$ | Foliar spray/soil application | *Lactuca sativa* | ↑fresh yield ↑dry biomass ↑root dry weight ↑plant nitrogen metabolism ↑Fv/Fm ↓oxidative stress ↑osmolytes ↑glucosynolates | [85] |

**Table 1.** *Cont.*

| ABIOTIC STRESS | SEVERITY AND TIME OF E×POUSURE | BIOSTIMULANT PRODUCT OR SUBSTANCES WITH A BIOSTIMULANT EFFECT | DOSE | APPLICATION METHODS AND NUMBER OF TREATMENTS | CROP | BENEFICIAL EFFECTS | REFERENCE |
|---|---|---|---|---|---|---|---|
| | 0.8, 1.3, and 1.8 dS/m NaCl | Retrosal® (organic mix with high concentration of carboxylic acids, containing calcium oxide (CaO) 8.0% (w/w) soluble in water and 1.4% complexed by ammonium ligninsulfonate, Zinc (Zn) 0.2% (w/w) soluble in water and 0.2% (w/w) chelated by EDTA.) | 0.1 or 0.2 mL/plant | Soil application (4×) | *Lactuca sativa* | ↑fresh weight ↑chlorophyll Pn ↑ gas exchange ↓proline ↓ABA | [162] |
| | 43, 207 mM NaCl for 7 weeks | *Achromobacter piechaudii* | - | Seedling inoculation | *Solanum lycopersicum* | ↑fresh and dry weights of tomato seedlings ↓ethylene ↑uptake phosphorous and potassium ↑WUE | [204] |
| | 200 mM NaCl | Nano-TiO$_2$ | 5, 10, 20 and 40 mg L$^{-1}$ | Foliar spray | *Solanum lycopersicum* | activities of carbonic anhydrase, nitrate reductase, SOD and POX ↑proline ↑glycinebetaine ↑growth ↑yield | [97] |
| | 28, 56 mmol kg$^{-1}$ | *Ascophyllum nodosum* | 1, 2 g kg$^{-1}$ | Soil application | *Cucumis sativus* | ↑fruit yield ↑Pn | [172] |
| | 7.15, 7.2 dSm$^{-1}$ | Licorice root extract | 0.50% | Seed soaking /foliar spray | *Phaseolus vulgaris* | ↑plant growth ↑yield ↑RWC ↑chlorophylls ↑free proline ↑total soluble carbohydrates ↑total soluble sugars ↑nutrients ↑selenium ↑K$^+$:Na$^+$ ratio ↑membrane stability index ↑activities of all enzymatic antioxidants ↓electrolyte leakage ↓MDA ↓Na$^+$ ↓H$_2$O$_2$ ↓O$_2^-$ | [181] |
| | 100 mM NaCl | Propolis and maize grain extract | 1, 2% | Soaking seed | *Phaseolus vulgaris* | ↑% germination ↑seedling growth ↑cell membrane stability index ↑RWC ↑free proline ↑total free amino acids ↑total soluble sugars ↑indole-3-acetic acid ↑gibberellic acid ↑activity of the antioxidant system ↓lipid peroxidation ↓electrolyte leakage ↓ABA | [178] |

**Table 1.** *Cont.*

| ABIOTIC STRESS | SEVERITY AND TIME OF EXPOUSURE | BIOSTIMULANT PRODUCT OR SUBSTANCES WITH A BIOSTIMULANT EFFECT | DOSE | APPLICATION METHODS AND NUMBER OF TREATMENTS | CROP | BENEFICIAL EFFECTS | REFERENCE |
|---|---|---|---|---|---|---|---|
| | 6.23–6.28 dS m$^{-1}$ | Salycilic acid and *Moringa oleifera* | 0.30% | Seed soaking /foliar spray | *Phaseolus vulgaris* | ↑shoot length ↑number and area of leaves ↑ plant dry weight ↑RWC ↑chlorophyll ↑carotenoid ↑total soluble sugars ↑free proline ↑ascorbic acid ↑N, P, K and Ca, ↑ratios of K/Na and Ca/Na ↑green pod and dry seed yields | [179] |
| | 100 mM NaCl | *Moringa oleifera* | crude extract | Soaking seed | *Phaseolus vulgaris* | ↑shoot and root lengths ↑plant dry mass ↑total soluble sugars ↑proline ↑K$^+$, Na$^+$ and Cl$^-$ ↑ascorbic acid ↑total glutathione ↓MDA ↓ H$_2$O$_2$ ↓O$_2^-$ ↑SOD, APX, GR | [177,180] |
| | 50, 150 mM NaCl | *Sargassum muticum* and *Jania rubens* | 1% | Foliar spray (2×) | *Cicer arietinum* | ↑plant growth ↑chlorophyll ↑carotenoid ↑soluble sugars ↑phenols ↓Na$^+$ ↑ K$^+$ ↓H$_2$O$_2$ ↑CAT, SOD, POD, APX activity ↓MDA | [176] |
| | 3, 6 g L$^{-1}$ | *Dunaliella salina* exopolysaccharides | 0.1 g L$^{-1}$ | Foliar spray (2×) | *Solanum lycopersicum* | ↑chlorophyll ↑protein ↓proline | [175] |
| | 8.81 dS m$^{-1}$ | Bee-honey based biostimulant | 25–50 g L$^{-1}$ | Foliar spray | *Allium cepa* | ↑biomass ↑bulb yield ↑WUE ↑photosynthetic pigments ↑osmoprotectants ↑membrane stability index ↑RWC ↑enzymatic and non-enzymatic antioxidants | [182] |
| | 8 mM NaCl | phosphorus / humic acid | 50, 100, 150 mg kg$^{-1}$ (P)/750, 1500 mg kg$^{-1}$ (humic acid) | Soil application | *Capsicum annuum* | ↑fresh and dry weight of shoot and root ↓membrane damage ↑nutrient uptake | [170] |
| UV-stress | 300–340nm illumination for 15 min | Nano-anatase | 0.25% | Soaking seed and foliar spray | *Spinacia oleracea* | ↓O$_2$ ↓H$_2$O$_2$ ↓MDA ↑ SOD, CAT, APX, GPX activity | [96] |

Fv/Fm maximum quantum efficiency of Photosystem II; Pn net photosynthetic rate; E transpiration rate; gs stomatal conductance; Ci sub stomatal CO$_2$ concentration; SLA specific leaf area; RGR relative growth rate; RLWC relative leaf water content; RWC relative water content; WUE water use efficiency; PI performance index; MDA malondialdehyde; TTC 2,3,5-triphenyltetrazolium chloride; GSH reduced glutathione; GSSG oxidized glutathione; LOX lipoxygenase; CAT catalase; SOD superoxide dismutase; APX ascorbate peroxidase; POX peroxidase; GR glutathione reductase; HI harvest index; ABA abscisic acid; ETR electron transport rate. The symbol ↑ means an increase or ↓ a decrease of the parameter measured. The symbol × represents how many times the treatment was applied.

**Author Contributions:** Conceptualization, R.B., G.F., A.F.; writing—original draft preparation, R.B. G.F.; writing—review and editing, A.F.

**Funding:** This research received no external funding.

**Conflicts of Interest:** The authors declare no conflict of interest.

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
