# Peer review of "Biostimulants Application in Horticultural Crops under Abiotic Stress Conditions"

_agronomy, doi:10.3390/agronomy9060306_

Round 1
Reviewer 1 Report
I read with interest the manuscript “Biostimulants application in horticultural crops under abiotic stress conditions”. The paper examine the literature about the use of biostimulants on vegetable crops to counteract environmental stresses.
My first comment relates to the fact that the paper is well written and interesting, also in light of the poor number of review on this topic and the significance of this work is high.
The absence of the line numbers has made the revision more difficult.
Abstract
1) “It has been observed recently that stress tolerance can be induced by biostimulants or specific bioactive compounds if they are applied when the plant really need to be protected”.
I suggest to rephrase the sentence because it is not clear and in part wrong (recently?????).
2) “Many companies are investing in the identification of the most effective bioactive molecules contained in different extracts or macerated raw materials, able to elicit specific plant responses against abiotic stresses”.
What do you mean with macerated???
2. Biostimulants
Biostimulants can contain traces of natural plant hormones, but their biological action should not be ascribed to them, otherwise they have to be registered as plant growth regulators.
Please delete this sentence because there are several papers reported that the effects of some biostimulants in plants metabolism is due to the ENDOGENOUS hormones contained into them.
2.1 Classification of biostimulants in categories
These positive effects could be mainly ascribed to the hormone-like activity (several hormones in the humus structure have been identified).
The hormone-like activity and the hormone activity are to very different concept!!!!! The hormone like activity is also due to the others compounds such us phenols or functional groups…….
Please rephrase this sentence.
I suggest to the authors to considered the paper: Microbiological features and bioactivity of a fermented manure product (Preparation 500) used in biodynamic agriculture, Journal of Microbiology and Biotechnology, 23, Issue 5, 3 April 2013, Pages 644-665.
Hydrolysed proteins and amino acids containing products: hydrolysed proteins are a mixture of amino acids, peptides, polypeptides and denatured protein that can be obtained by chemical, enzymatic and thermal hydrolysis of proteins from both plant and animal sources [52,62].
Several biostimulants were produced also by a mix of chemical, enzymatic and thermal hydrolysis.
Author Response
I read with interest the manuscript “Biostimulants application in horticultural crops under abiotic stress conditions”. The paper examine the literature about the use of biostimulants on vegetable crops to counteract environmental stresses.
My first comment relates to the fact that the paper is well written and interesting, also in light of the poor number of review on this topic and the significance of this work is high.
The absence of the line numbers has made the revision more difficult.
Author answer (A.A.): Thank you for the positive evaluation of our review paper. We apologies for missing the rows numbers that now are indicated.
Abstract
1) “It has been observed recently that stress tolerance can be induced by biostimulants or specific bioactive compounds if they are applied when the plant really need to be protected”.
I suggest to rephrase the sentence because it is not clear and in part wrong (recently?????).
A.A.: As suggested the sentence has been revised.
2) “Many companies are investing in the identification of the most effective bioactive molecules contained in different extracts or macerated raw materials, able to elicit specific plant responses against abiotic stresses”.
What do you mean with macerated???
A.A.: Thank you for the correction. Actually, maceration is one of the possible extractive techniques, so we decided to keep in the sentence only the word “extracts” that includes all the extraction methods.
2. Biostimulants
Biostimulants can contain traces of natural plant hormones, but their biological action should not be ascribed to them, otherwise they have to be registered as plant growth regulators.
Please delete this sentence because there are several papers reported that the effects of some biostimulants in plants metabolism is due to the ENDOGENOUS hormones contained into them.
A.A.: As suggested we deleted this sentence.
2.1 Classification of biostimulants in categories
These positive effects could be mainly ascribed to the hormone-like activity (several hormones in the humus structure have been identified).
The hormone-like activity and the hormone activity are to very different concept!!!!! The hormone like activity is also due to the others compounds such us phenols or functional groups…….
Please rephrase this sentence.
I suggest to the authors to considered the paper: Microbiological features and bioactivity of a fermented manure product (Preparation 500) used in biodynamic agriculture, Journal of Microbiology and Biotechnology, 23, Issue 5, 3 April 2013, Pages 644-665.
Hydrolysed proteins and amino acids containing products: hydrolysed proteins are a mixture of amino acids, peptides, polypeptides and denatured protein that can be obtained by chemical, enzymatic and thermal hydrolysis of proteins from both plant and animal sources [52,62].
Several biostimulants were produced also by a mix of chemical, enzymatic and thermal hydrolysis.
A.A.: As suggested, the sentence and the literature have been revised. The biodynamic preparations are not considered in the biostimulants classification, if even they can contain bioactive compounds. Moreover, the action of biodynamic preparation is not only related to their composition but also to specific procedure in their preparation.
Reviewer 2 Report
I read with interest the review entitled ”Biostimulants application in horticultural crops under abiotic stress conditions”.
This review analyzes the recent results obtained with the use of biostimulant products, with particular regard to their effects that improve the tolerance to abiotic stresses in vegetable crops.
After a brief analysis of the different biostimulants, their effects on different types of abiotic stress (cold stress, salinity, drought and nutrient deficiency) are presented. To facilitate the reading of the review, an interesting table is presented, in which examples of application of products based on biostimulants to overcome various abiotic stresses are reported. The authors rightly point out that it is difficult to understand and explain the effects that the application of biostimulants determines, because each biostimulant is composed of different molecules, each with its own metabolic target,” Therefore they could be classified on the basis of their role in plants”.
The review is well written and well set, it presents a considerable number of references (191) of which 9, refer to the period 1979-2000 and all the others published in the last 19 years.
In my experience, the only weak point I highlight is that the authors never take advantage of the explanation of the difficult mechanisms that underlie the action of biostimulants. In fact, while the effect on growth and morphology of plants treated with biostimulants is well described, the descriptions of the mechanisms could be improved.
Author Response
I read with interest the review entitled ”Biostimulants application in horticultural crops under abiotic stress conditions”.
This review analyzes the recent results obtained with the use of biostimulant products, with particular regard to their effects that improve the tolerance to abiotic stresses in vegetable crops.
After a brief analysis of the different biostimulants, their effects on different types of abiotic stress (cold stress, salinity, drought and nutrient deficiency) are presented. To facilitate the reading of the review, an interesting table is presented, in which examples of application of products based on biostimulants to overcome various abiotic stresses are reported. The authors rightly point out that it is difficult to understand and explain the effects that the application of biostimulants determines, because each biostimulant is composed of different molecules, each with its own metabolic target,” Therefore they could be classified on the basis of their role in plants”.
The review is well written and well set, it presents a considerable number of references (191) of which 9, refer to the period 1979-2000 and all the others published in the last 19 years.
In my experience, the only weak point I highlight is that the authors never take advantage of the explanation of the difficult mechanisms that underlie the action of biostimulants. In fact, while the effect on growth and morphology of plants treated with biostimulants is well described, the descriptions of the mechanisms could be improved.
A.A.: Thank you for the positive comments on our work. Considering the final suggestion, we would like to specify that the difficulty in determining the mechanism of action of biostimulants is one of main problem of this sector. At present, the mechanism of action of biostimulant products is largely unknown, this because of the complex nature of the raw materials used for their production and the heterogeneous mixture of components present in the final product. This aspect is reported in the paragraph 2 and in the conclusions section. However, when possible, we tried to give probable explanations concerning their mode of action or we reported what has been hypothesized by other authors in their works.
Reviewer 3 Report
This review give a complete overview on the various effects of biostimulants on horticultural crops under abiotic stress conditions.
A few changes (listed below) should be made to improve the readability of this paper.
Chapter 2: Biostimulants
This chapter should be better organized and extended. The issue concerning biostimulants from the regulatory aspects to their mechanisms of action on plant physiology and on plant-rizosphere system is very important and it is preliminary to all the subsequent paragraphs. A number of paper on this matter are not cited and must be included. Furthermore, some sentences seem a little disconnected one each other, as for example the first five lines of page 4 (“Biostimulants products have been considered…”), that is correct but it could be moved elsewhere in the text (may be before?).
And again, in the middle of the paragraph (“The complex nature of…”) a very important subject is faced, but in the present version it appears isolated from the context and it needs to be better integrated in the text and also deepened.
Pag.4: I suggest to remove “or hormones” from this sentences “Mineral elements, vitamins, amino acids, chitin, chitosan, and poly- and oligosaccharides are the most ordinary components, but the biostimulant activity does not depend on the product’s nutrient or hormones content exclusively”, because in general bisotimulants should not contain hormones.
Pag. 4: Please modify this sentence: “The new EU Fertilising Products Regulation has been approved in October 2017 while the Compromise on the Fertilising Products Regulation (FPR) on Wednesday has been approved in March 2019 [47,48]” with the following: “The amendments to the proposal of the European Commission were adopted by the European Parliament in October 2017 while the legislative resolution on the proposal was approved on 27 March 2019 [47,48].”
Pag. 5, in the middle: It is important to try to better deepen and explain the effects of biostimulants on PAL and the importance of this enzyme and the physiological and agronomical relevance of these results. If it is not possible, please eliminate this part.
Paragraph 2.3. The first four lines are a repetition of concepts already stated, Please remove or synthetize.
In general, especially in this introductory section on biostimulants (Chapter 2) and on their classification (2.1) the bibliography cited should be deeply revised. I found a lot of inappropriate or out of context citations.
Few example of this are listed below: the citation 37-38-39 which should be removed (or moved elsewhere in relation to different and more specific contexts) and substituted with others, as for example one or more review (the 38 might be shifted to the section dealing with salinity stress). And again the 40 and 41, even if are very good, are not dealing with the omic technologies.
Moreover, as far as humic substances effects and their hormonal effects are concerned (51-52-53-54) few additional very important papers should be cited, as for example Canellas et al. 2002, Zandonaldi et al, 2007, 2010, Trevisan et al 2010, 2011, Quaggiotti et al 2004 and others.
Finally two recent paper on omics approach on the study of biostimulants should also be included: Zamboni et al Front Plant Sci. 2017 and Trevisan et al, J Agric Food Chem. 2017.
When describing the biostimulant effects on the stress response please refers also to Trevisan et al (Agronomy, 2019).
Finally, the last chapter (Conclusions and future prospects) should be extensively widened. Please, try also to make it clearer what do you exactly mean with the two last sentences.
Furthermore the English text is sometimes improper and should benefit by a revision by a native English speaker.
Please check accurately all the manuscript because I found many editing errors.
Author Response
This review give a complete overview on the various effects of biostimulants on horticultural crops under abiotic stress conditions.
A few changes (listed below) should be made to improve the readability of this paper.
Chapter 2: Biostimulants
This chapter should be better organized and extended. The issue concerning biostimulants from the regulatory aspects to their mechanisms of action on plant physiology and on plant-rizosphere system is very important and it is preliminary to all the subsequent paragraphs. A number of paper on this matter are not cited and must be included. Furthermore, some sentences seem a little disconnected one each other, as for example the first five lines of page 4 (“Biostimulants products have been considered…”), that is correct but it could be moved elsewhere in the text (may be before?).
A.A.: Thanks for the suggestions. We revised and modified the chapter 2.
And again, in the middle of the paragraph (“The complex nature of…”) a very important subject is faced, but in the present version it appears isolated from the context and it needs to be better integrated in the text and also deepened.
A.A.: We tried to explain this important aspect related to biostimulants also in other sections of the manuscript, in order to give it more importance.
Pag.4: I suggest to remove “or hormones” from this sentences “Mineral elements, vitamins, amino acids, chitin, chitosan, and poly- and oligosaccharides are the most ordinary components, but the biostimulant activity does not depend on the product’s nutrient or hormones content exclusively”, because in general bisotimulants should not contain hormones.
A.A.: From a legal point of view, biostimulants can contain traces of natural plant hormones, but their biological action should not be ascribed to them, otherwise they should be registered as plant growth regulators, as reported by several authors (Bulgari et al., 2015; Du Jardin, 2015; Yakhin et al., 2017). For this reason, we specified that “natural plant hormones” can be part of the compositions. The text has been revised considering also the comments of the reviewer 1 on this specific issue.
Pag. 4: Please modify this sentence: “The new EU Fertilising Products Regulation has been approved in October 2017 while the Compromise on the Fertilising Products Regulation (FPR) on Wednesday has been approved in March 2019 [47,48]” with the following: “The amendments to the proposal of the European Commission were adopted by the European Parliament in October 2017 while the legislative resolution on the proposal was approved on 27 March 2019 [47,48].”
A.A.: As suggested the sentence has been replaced.
Pag. 5, in the middle: It is important to try to better deepen and explain the effects of biostimulants on PAL and the importance of this enzyme and the physiological and agronomical relevance of these results. If it is not possible, please eliminate this part.
A.A.: We added in the text a sentence related to biostimulants effect on this enzyme.
Paragraph 2.3. The first four lines are a repetition of concepts already stated, Please remove or synthetize.
A.A.: As suggested, the sentence has been shortened.
In general, especially in this introductory section on biostimulants (Chapter 2) and on their classification (2.1) the bibliography cited should be deeply revised. I found a lot of inappropriate or out of context citations.
A.A.: The chapter 2 and citations have been rechecked.
Few example of this are listed below: the citation 37-38-39 which should be removed (or moved elsewhere in relation to different and more specific contexts) and substituted with others, as for example one or more review (the 38 might be shifted to the section dealing with salinity stress). And again the 40 and 41, even if are very good, are not dealing with the omic technologies.
A.A.: Citations have been rechecked and modified considering the reviewer comments
Moreover, as far as humic substances effects and their hormonal effects are concerned (51-52-53-54) few additional very important papers should be cited, as for example Canellas et al. 2002, Zandonaldi et al, 2007, 2010, Trevisan et al 2010, 2011, Quaggiotti et al 2004 and others.
A.A.: The suggested papers have been added to the manuscript
Finally two recent paper on omics approach on the study of biostimulants should also be included: Zamboni et al Front Plant Sci. 2017 and Trevisan et al, J Agric Food Chem. 2017.
A.A.: The papers have been considering in the revised text and properly cited.
When describing the biostimulant effects on the stress response please refers also to Trevisan et al (Agronomy, 2019).
A.A.: As suggested, the paper has been added to chapter 2.1.
Finally, the last chapter (Conclusions and future prospects) should be extensively widened. Please, try also to make it clearer what do you exactly mean with the two last sentences.
A.A.: This section has been revised.
Furthermore the English text is sometimes improper and should benefit by a revision by a native English speaker.
A.A.: the English has been revised according the suggestion.
Please check accurately all the manuscript because I found many editing errors.
A.A.: We have now carefully revised all the manuscript in view of the reviewer comments.
Reviewer 4 Report
The review “Biostimulants application in horticultural crops under abiotic stress conditions” is an interesting elaboration, which fits into the main strategies of sustainable and ecological plant production. The use of biostimulants in horticultural crops can help to avoid main problems of modern agriculture: the intensive use of agricultural chemicals, soil degradation, low biodiversity of agricultural ecosystems, low biological quality of the yield, etc. In my opinion, the report on the progress of biostimulant market as well as biostimulant application in horticultural crops is an important contribution to the development of horticultural science. However, I have some issues which should be addressed before the publication of the paper.
- The novelty of this study is not revealed, even in the crucial parts (Abstract, Conclusion). In the present form, it is just another elaboration of a well-known topic.
- Authors pointed (Chapter 1) that there are many strategies applied to improve stress tolerance in horticultural plants (genetic, agronomical) but only genetic improvement and grafting were described. In my opinion, in vitro selection and the most important agronomic strategies should be at least mentioned as a background for biostimulant application.
- Subchapter 2.3 contains the simple compilation of the results on biostimulant action in conditions of particular abiotic stresses. It will be valuable if each subchapter is ended with a few sentences concluding on the major point, trend or result that Authors have deduced from literature analysis.
- In my opinion, it should be underlined that all abiotic stress factors lead to the overproduction of reactive oxygen species, and have the same common signal and responding pathways in plants. Moreover, stress always occurs as a complex of various interacting environmental factors that contribute to varying degrees to the overall plant status. Mentioned aspects deserve to be covered in greater depth. It will be valuable to compare the action of particular biostimulant (or biostimulant category) in conditions of different/combined abiotic stresses (in the form of a table, etc).
- Some biostimulants (seaweed extracts, for example) are a direct source of nutrients - this issue should be also included in 2.3.5 chapter.
- Keywords should not repeat words from the title.
- Page 2 “Environmental stresses generally cause oxidative damage in plants...” - This paragraph is over-generalization. Moderate stress can accelerate ROS scavengers and other secondary metabolites synthesis leading to increase of the nutritional value of crops.
Author Response
The review “Biostimulants application in horticultural crops under abiotic stress conditions” is an interesting elaboration, which fits into the main strategies of sustainable and ecological plant production. The use of biostimulants in horticultural crops can help to avoid main problems of modern agriculture: the intensive use of agricultural chemicals, soil degradation, low biodiversity of agricultural ecosystems, low biological quality of the yield, etc. In my opinion, the report on the progress of biostimulant market as well as biostimulant application in horticultural crops is an important contribution to the development of horticultural science. However, I have some issues which should be addressed before the publication of the paper.
- The novelty of this study is not revealed, even in the crucial parts (Abstract, Conclusion). In the present form, it is just another elaboration of a well-known topic.
A.A.: The aim of this review is to show the state of the art of the use of biostimulants on vegetable crops, focusing the attention on their capability to counteract environmental stresses. At present, in literature there are few similar works (review). Moreover, we believe that both abiotic stresses and biostimulants are two very important topics for improving the agriculture sustainability. Crops are in fact subjected to abiotic stresses during their growing cycle, that could greatly reduce the productivity and quality of crops. Biostimulants could represent an effective and sustainable tool to employ in crop management to enhance plant growth and productivity, improving also the defense mechanisms. We have better clarified these aspects in the conclusions.
Authors pointed (Chapter 1) that there are many strategies applied to improve stress tolerance in horticultural plants (genetic, agronomical) but only genetic improvement and grafting were described. In my opinion, in vitro selection and the most important agronomic strategies should be at least mentioned as a background for biostimulant application.
A.A.: As suggested, we added in vitro selection and some agronomical strategy in this chapter.
Subchapter 2.3 contains the simple compilation of the results on biostimulant action in conditions of particular abiotic stresses. It will be valuable if each subchapter is ended with a few sentences concluding on the major point, trend or result that Authors have deduced from literature analysis.
A.A.: Thank you for your suggestion, we added as suggested few sentences at the end of section the possible effect of the biostimulant in protecting the crops against the specific stress. However these information are general because many information are missing for each particular abiotic stress and further researches are needed to elucidate the specific action of the biostimulant for the specific stress.In my opinion, it should be underlined that all abiotic stress factors lead to the overproduction of reactive oxygen species, and have the same common signal and responding pathways in plants. Moreover, stress always occurs as a complex of various interacting environmental factors that contribute to varying degrees to the overall plant status. Mentioned aspects deserve to be covered in greater depth. It will be valuable to compare the action of particular biostimulant (or biostimulant category) in conditions of different/combined abiotic stresses (in the form of a table, etc).
A.A.: Thanks for your comment. We added a part related to ROS in the text of the paragraph 1.
Some biostimulants (seaweed extracts, for example) are a direct source of nutrients - this issue should be also included in 2.3.5 chapter.
A.A.: Thank you for your suggestion. Even knowing that seaweed extract, for example, could be a direct source of minerals and nutrients for plants. As biostimulant definition the minerals content should have no effect on plant growth, but the improved performance of the plants must be determined by the organic components of the products. If the mineral components significantly increase the plant performance the products is not a biostimulant but a fertilizer. However, this aspect has been discussed in the review.
Keywords should not repeat words from the title.
A.A.: As suggested, the keywords have been checked and modified.
Page 2 “Environmental stresses generally cause oxidative damage in plants...” - This paragraph is over-generalization. Moderate stress can accelerate ROS scavengers and other secondary metabolites synthesis leading to increase of the nutritional value of crops.
A.A.: Thank you for the correction. Few sentences have been added to the manuscript to better explain this idea. However, we would like to emphasize that the topic of the work is the application of biostimulant products to counteract abiotic stresses when they are detrimental/dangerous for plants and not when these kind of stresses positively affect crops quality.
Round 2
Reviewer 4 Report
The manuscript was significantly improved and I found the only minor, editing errors, so I recommend to accept after technical correction.
Some errors, I've noticed, are pointed in the attached file.

Author Response
Thank you for your comments, they were really useful to improve the quality of the manuscript.
As suggested we made some corrections and fixed the edting errors.
Thank you for the remark (comment line 155). We slightly changed the sentences in order to better clarify this aspect and focus the attention especially on the differences between cultivars. This is a broad and complex topic and differences in antioxidant capacity could be found both in species, CV or even individual genotypes, as you correctly said. For this reason we used the word "plants" in the previous version of the manuscript with the intention of include these cases as a whole. We hope that the speech will be more clear and smooth after the corrections.